# Wavelength-tunable high-fidelity entangled photon sources enabled by dual Stark effects

Chen Chen[1,10], Jun-Yong Yan [1,10], Hans-Georg Babin[2], Jiefei Wang[3], Xingqi Xu[3], Xing Lin [1], Qianqian Yu[4], Wei Fang [5], Run-Ze Liu [6], Yong-Heng Huo [6], Han Cai [5], Wei E. I. Sha [1], Jiaxiang Zhang [7], Christian Heyn[8], Andreas D. Wieck [2], Arne Ludwig [2], Da-Wei Wang [3,9], Chao-Yuan Jin[1] & Feng Liu [1] ✉

The construction of a large-scale quantum internet requires quantum repeaters containing multiple entangled photon sources with identical wavelengths. Semiconductor quantum dots can generate entangled photon pairs deterministically with high fidelity. However, realizing wavelength-matched quantum-dot entangled photon sources faces two difficulties: the non-uniformity of emission wavelength and exciton fine-structure splitting induced fidelity reduction. Typically, these two factors are not independently tunable, making it challenging to achieve simultaneous improvement. In this work, we demonstrate wavelength-tunable entangled photon sources based on droplet-etched GaAs quantum dots through the combined use of AC and quantum-confined Stark effects. The emission wavelength can be tuned by ~1 meV while preserving an entanglement fidelity $f$ exceeding 0.955(1) in the entire tuning range. Based on this hybrid tuning scheme, we finally demonstrate multiple wavelength-matched entangled photon sources with $f > 0.919(3)$, paving the way towards robust and scalable on-demand entangled photon sources for quantum internet and integrated quantum optical circuits.

The quantum internet is a network capable of transmitting qubits and connecting multiple quantum processors[1]. Such quantum networks are the foundation of numerous quantum information technologies, particularly for distributed quantum computing[2] and quantum communications[3]. Expanding the range of the quantum internet to a global scale requires quantum repeaters[4] consisting of multiple entangled photon sources (EPSs) with identical emission wavelengths. A variety of systems can generate entangled photons, such as nonlinear

crystals[5], waveguides[6], micro-resonators[7], and cold atoms[8]. Among those candidates, semiconductor quantum dots (QDs) are one of the most promising platforms to generate polarization-entangled photon pairs via biexciton cascade decay (see Fig. 1) with advantages of electrical controllability[9,10], on-demand operation[11], high brightness[12], and near-unity entanglement fidelity[13]. Furthermore, entanglement swapping - a basic operation of quantum repeaters - has been demonstrated with time-multiplexed entangled photon pairs emitted from a single QD[14,15].

[1]State Key Laboratory of Extreme Photonics and Instrumentation, College of Information Science and Electronic Engineering, Zhejiang University, Hangzhou 310027, China. [2]Lehrstuhl für Angewandte Festkörperphysik, Ruhr-Universität Bochum, 44801 Bochum, Germany. [3]Zhejiang Province Key Laboratory of Quantum Technology and Device, School of Physics, Zhejiang University, Hangzhou 310027, China. [4]Zhejiang Laboratory, Hangzhou 311100, China. [5]College of Optical Science and Engineering, Zhejiang University, Hangzhou 310027, China. [6]Hefei National Research Center for Physical Sciences at the Microscale and School of Physical Sciences, University of Science and Technology of China, Hefei 230026, China. [7]National Key Laboratory of Materials for Integrated Circuits, Shanghai Institute of Microsystem and Information Technology, Chinese Academy of Sciences, Shanghai 200050, China. [8]Center for Hybrid Nanostructures (CHyN), University of Hamburg, Luruper Chaussee 149, 22761 Hamburg, Germany. [9]CAS Center for Excellence in Topological Quantum Computation, University of Chinese Academy of Sciences, Bejing 100190, China. [10]These authors contributed equally: Chen Chen, Jun-Yong Yan. ✉e-mail: feng_liu@zju.edu.cn

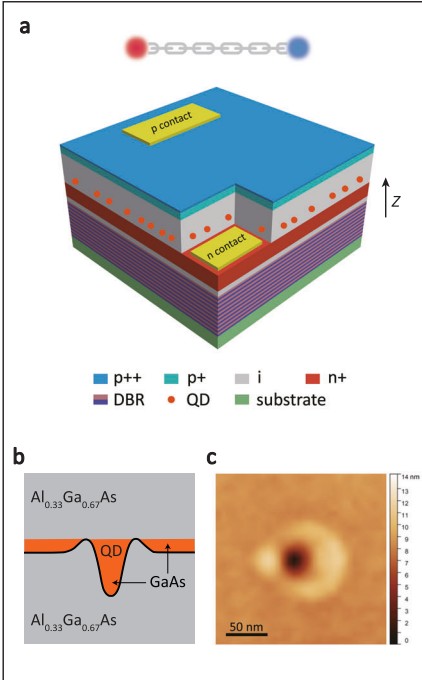

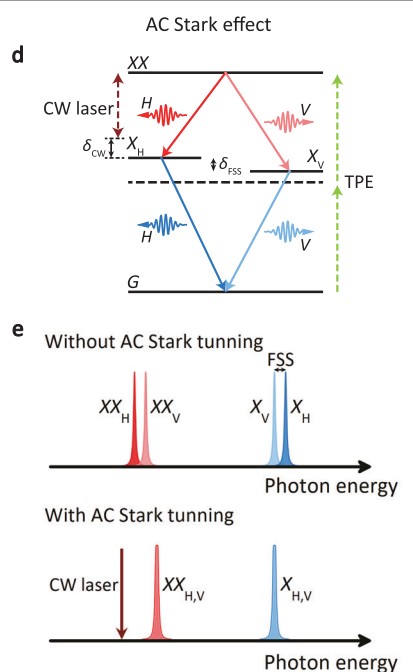

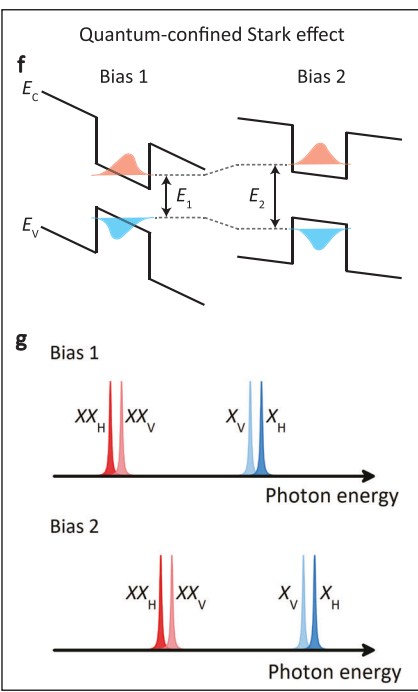

**Fig. 1 | Device and hybrid tuning scheme. a** Device structure. The device consists of a distributed Bragg reflector (DBR) and an n-i-p diode structure with droplet-etched GaAs QDs. **b** Cross-section of GaAs QD formed in $Al_{0.33}Ga_{0.67}As$ bulk material. **c** Atomic force microscopy (AFM) image of a droplet-etched GaAs QD. **d** AC Stark effect and QD energy level diagram. A CW laser (brown arrow) that is H-polarized and red-detuned by $\delta_{CW}$ from $|XX\rangle \rightarrow |X_H\rangle$ transition shifts the $XX$ and $X_H$ energies while leaving $X_V$ unchanged. Polarization-entangled photon pairs can be generated via biexciton (*XX*)-exciton (*X*) cascade decay process. The *XX* state can be deterministically prepared by resonant two-photon excitation (TPE, green arrow). Two single-exciton eigenstates ($|X_H\rangle$ and $|X_V\rangle$) are split by $\delta_{FSS}$. **e** Schematic of fluorescence spectra with and without AC Stark tuning under TPE excitation. **f** Wavelength tuning by quantum-confined Stark effect. Changing the bias results in different energy band bending, which causes a shift of the electron (red) and hole (blue) wavefunction, thus changing the transition energy. **g** Schematic of fluorescence spectra in the presence of FSS at different biases.

However, building true quantum repeaters requires multiple wavelength-matched QD EPSs, which remains challenging due to the non-uniformity of QD emission wavelength and exciton (*X*) fine-structure splitting (FSS) causing imperfect entanglement fidelity[16,17]. The former originates from the inhomogeneous QD size, shape, composition, and strain[18,19]. The latter is caused by the anisotropy of QDs[20] which lifts the degeneracy of two single-exciton eigenstates ($X_H/X_V$) via electron-hole exchange interaction[16].

In order to realize QD EPSs with identical wavelengths and high entanglement fidelity, considerable efforts have been made to tune the exciton energy and FSS by applying strain[21,22], magnetic field[23,24], optical field[25,26], and static electric field[27,28]. Wavelength tuning range up to 25 meV[29] and close-to-zero FSS[21,24] have been demonstrated separately. However, since the control knobs for these two parameters are shared, tuning one of them inevitably affects the other. Therefore, one tuning knob is typically not sufficient to shift the QD to the desired wavelength while keeping the FSS negligible. To tackle this challenge, more advanced tuning schemes based on multi-axis strain/electric fields[30,31] or involving multiple tuning mechanisms, e.g. magnetic field and electric field[32], have been developed. Simultaneous tuning of QD emission wavelength and FSS has been realized[30,33]. Furthermore, under multi-axis strain tuning, near-unity entanglement fidelity (97.8%) has been achieved at a fixed wavelength[13]. Although significant progress has been made, a wavelength-tunable QD EPS with high entanglement fidelity across the entire tuning range has not been demonstrated.

In this work, we present a wavelength-tunable QD EPS with an entanglement fidelity exceeding 0.955(1) across the entire tuning range. This is achieved through a hybrid scheme simultaneously tuning the QD emission wavelength and exciton FSS via the quantum-confined Stark effect and AC Stark effect, respectively. The wavelength tuning range (~1 meV) is two orders of magnitude larger than the QD emission linewidth. The performance of the hybrid tuning scheme and our device is further examined according to requirements of practical applications. The stability of the QD EPS is confirmed by maintaining high entanglement fidelity over a long period of time without re-adjustment of the bias and laser power. The scalability of this hybrid tuning scheme is verified by performing experiments on different QDs. Up to 39 QDs can be tuned to the same emission wavelength. Finally, we demonstrate high entanglement fidelity ($f > 0.919(3)$) for multiple QDs tuned into resonance with each other or with Rb atoms. This work provides a viable approach towards robust and scalable wavelength-tunable EPSs.

## Results

### Device and hybrid tuning scheme

Our EPS device consists of droplet-etched GaAs QDs embedded in an n-i-p diode[34] (shown in Fig. 1a, b). The slightly asymmetric shape (see the AFM image in Fig. 1c) of QDs results in a finite exciton FSS via the electron-hole exchange interaction[16]. DC electric field along the QD growth direction (Z direction) can be applied across QDs by biasing the n-i-p diode. To ensure optimum performance, the device is operated at 3.6 K to minimize the dephasing[35] and coupling between excitonic states[36] caused by phonons.

Polarization-entangled photon pairs can be generated from a single GaAs QD via biexciton cascade decay (see Fig. 1d). In this process, two electron-hole pairs (biexciton, $|XX\rangle$) are initially created by simultaneously absorbing two photons. Then the two electron-hole pairs recombine successively and emit a pair of non-degenerate photons separated by the biexciton binding energy through two possible

decay channels. In the ideal case without FSS, the two photons are maximally polarization-entangled[17,37,38]:

$$|\Phi^+\rangle = \frac{1}{\sqrt{2}}(|H_{XX}\rangle|H_X\rangle + |V_{XX}\rangle|V_X\rangle), \qquad (1)$$

where $|H_{XX}/V_{XX}\rangle$ denotes the biexciton photon emitted via $|XX\rangle \rightarrow |X_{H/V}\rangle$ transition with horizontal/vertical polarization. $|H_X/V_X\rangle$ denotes the exciton photon emitted via $|X_{H/V}\rangle \rightarrow |G\rangle$ transition. However, in reality, two intermediate single-exciton eigenstates $(|X_{H/V}\rangle)$ have a finite splitting $\hbar\delta_{FSS}$ in most QDs. This FSS results in a reduced time-integrated entanglement fidelity $f$ according to[17,38]:

$$f = \frac{1}{4}\left(2 - g^{(2)}(0) + \frac{2(1 - g^{(2)}(0))}{1 + (\delta_{FSS}\tau_X/\hbar)^2}\right), \qquad (2)$$

where $g^{(2)}(0)$ and $\tau_X$ denote second-order correlation function at zero delay and exciton lifetime, respectively. Other dephasing mechanisms, such as spin scattering and cross-dephasing between single-exciton eigenstates are not considered here[17].

Therefore, in order to realize a wavelength-tunable QD EPS with high entanglement fidelity, it is crucial to tune $X_{H/V}$ energy and simultaneously eliminate the exciton FSS. To this end, we propose a hybrid tuning scheme. The wavelength is tuned by the quantum-confined Stark effect[39], where the DC electric field tilts the band structure, resulting in a change of the energy difference between the electron and hole (see Fig. 1f), hence a shift of the QD emission wavelength (see Fig. 1g). The change of all transition energies with DC electric field is described by[39]:

$$E = E_0 - p_z F_z + \beta F_z^2, \qquad (3)$$

where $p_z$ and $F_z$ represent the permanent dipole moment and applied DC electric field, respectively. $\beta$ is the polarizability.

After tuning the QD to the desired emission wavelength, the exciton FSS can then be eliminated via the AC Stark effect[25,40-43], where a linearly, e.g. horizontally, polarized CW laser slightly red-detuned from $|XX\rangle \rightarrow |X_H\rangle$ transition shifts $X_H$ energy while leaving the cross-polarized $X_V$ state unchanged (see Fig. 1d). Here, the directions of horizontal and vertical polarizations are determined by the two single-exciton eigenstates. By properly choosing the power and detuning of the CW laser, the FSS can be reduced to zero (see Fig. 1e). The change in FSS ($\Delta\omega$) with CW laser power and detuning is given by ref. 26:

$$\Delta\omega = \frac{\delta_{CW}}{2}\left(1 - \frac{\sqrt{\Omega_{CW}^2 + \delta_{CW}^2}}{|\delta_{CW}|}\right), \qquad (4)$$

where $\Omega_{CW}$ is the Rabi frequency, proportional to the square root of the laser power. $\delta_{CW} = E_{XX}/\hbar - \omega_{CW}$ is the detuning of the CW laser relative to $|XX\rangle \rightarrow |X_H\rangle$ transition.

We note that the AC Stark effect is an universal method to tune the FSS of arbitrary QDs to zero[25]. Therefore even though the DC electrical field required by the wavelength tuning may affect $\delta_{FSS}$ and polarization directions of two single-exciton eigenstates[27], the FSS can still be completely compensated at different bias by making full use of the CW laser's degrees of freedom including power, detuning and polarization.

## Simultaneous tuning of wavelength and FSS
Before we move to the measurement of the entanglement fidelity, the first key step is to show the simultaneous tuning of QD emission wavelength and exciton FSS. The tuning of the exciton/biexciton emission wavelength is demonstrated by sweeping the bias of the n-i-p diode and measuring the fluorescence spectra under resonant two-photon excitation (green dashed arrows in Fig. 1d) with a pulse area of $\Theta = \pi$. The pulse area $\Theta$ is calibrated by performing a two-photon Rabi oscillation measurement (see Supplementary Fig. 2). The $X$ ($XX$) photon energy is tuned by 1.08 meV (0.76 meV) with increase of the bias from 0 V to 0.25 V (see Fig. 2a). This tuning range is two orders of magnitude larger than the QD emission linewidth (5.37 $\mu$eV, see Supplementary Fig. 3a). In the rest of the paper, we refer to this QD as QD A.

Next, we demonstrate the elimination of FSS at different QD emission wavelengths. To measure the FSS, the QD emission is sent through a rotatable half-wave plate (HWP) and a fixed linear polarizer. The FSS can be extracted by fitting the energy difference between $XX$ and $X$ fluorescence peaks as a function of the HWP angle with a sinusoidal function (see Fig. 2b)[44]. QD A shows an intrinsic FSS of 2.92(7) $\mu$eV. The HWP angles where the energy difference reaches the minimum/maximum correspond to horizontal and vertical polarization directions determined by two single-exciton eigenstates, respectively. The FSS can be compensated via the AC Stark effect by illuminating the QD with a horizontally polarized CW laser red-detuned by 303 $\mu$eV from $|XX\rangle \rightarrow |X_H\rangle$ transition (see brown dashed arrow in Fig. 1d). Figure 2c–h show the elimination of the FSS by sweeping the CW laser power. The FSS can be reduced to almost zero at different biases which correspond to different $X/XX$ emission wavelengths indicated by colored arrows in Fig. 2a. These results prove that our hybrid tuning scheme is able to tune the QD emission wavelength while keeping the FSS close to zero.

## Wavelength-tunable EPS with high entanglement fidelity
We have shown that the precondition for realizing a wavelength-tunable QD EPS has been fulfilled with our hybrid tuning scheme. However, many open questions still need to be answered before we can claim that we have a wavelength-tunable EPS. For example, it is unclear whether the entanglement fidelity will be degraded by the charge noise created by the CW-laser-generated charge carriers. To clarify these questions, it is necessary to directly examine the entanglement fidelity at different QD emission wavelengths.

In order to evaluate the entanglement fidelity $f$ with respect to the state $|\Phi^+\rangle$, quantum tomography experiment[45,46] is performed. $f$ is extracted from the two-photon density matrix constructed from 16 cross-correlation measurements between $X$ and $XX$ photons (see details in Methods). Under this condition, $f$ reaches 0.952(1) at $V_g = 0.1$ V and a CW laser power of 14.6 $\mu$W (see Fig. 3a). As shown in Fig. 3b, we also evaluate $f$ using a reduced measurement set involving 6 correlation measurements (see Supplementary Note 4). An entanglement fidelity $f = 0.957(1)$ is obtained, in good agreement with that extracted from full quantum tomography. The errors of entanglement fidelity are estimated using error propagation method assuming a Poisson statistics of the coincidence counts[13,47]. For the sake of simplicity, we measure $f$ with the reduced measurement set in the rest of this paper.

Figure 3 c shows $f$ measured at different $X$ photon energies. $f$ with AC Stark tuning remains above 0.95 in the entire 1.08 meV tuning range (red squares). The slight increase of $f$ at the edges of the charge plateau is attributed to the decreased $X$ lifetime ($\tau_X$) caused by co-tunnelling (see Supplementary Fig. 4)[48]. As a comparison, $f$ without AC Stark tuning is around 0.7 (blue squares). These results prove that our hybrid tuning scheme indeed provides a viable approach toward a wavelength-tunable QD EPS with high entanglement fidelity.

## Stability
In order to support a practical quantum network, it is essential that the EPS can be continuously operated for a long period of time, ideally without the concern of periodic re-adjustment of the bias and laser

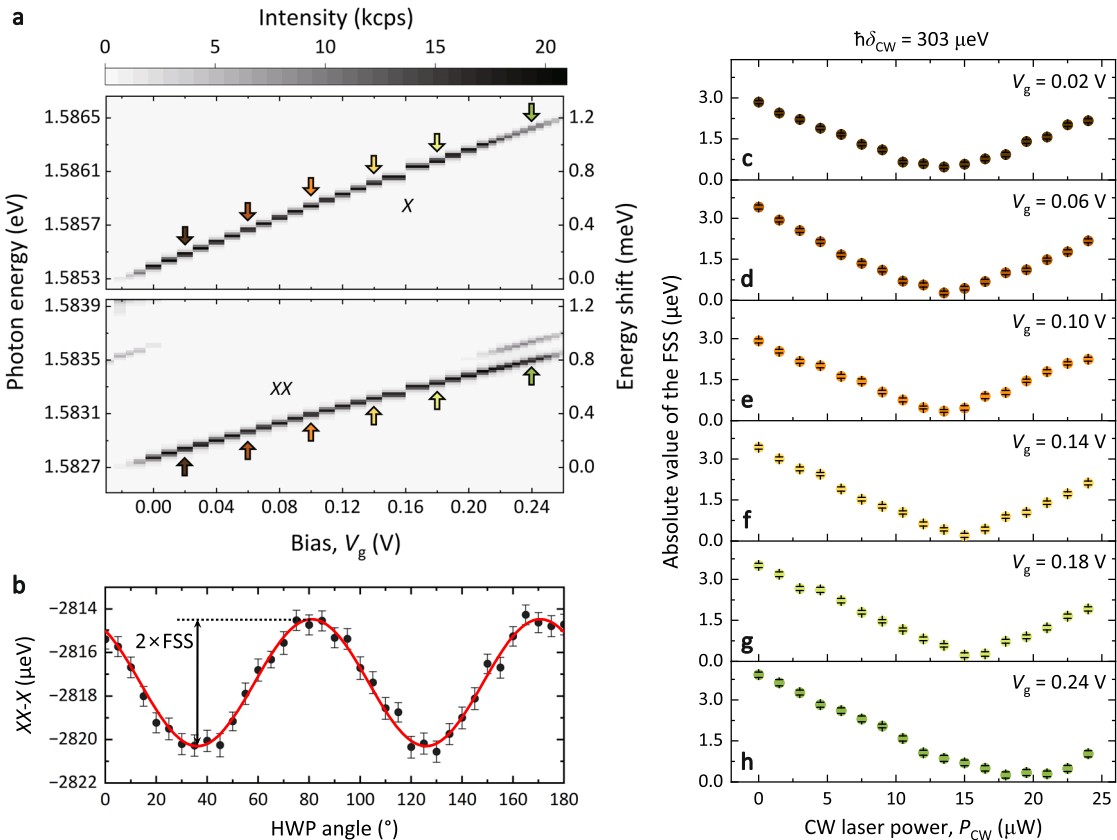

**Fig. 2 | Simultaneous tuning of photon energy and fine-structure splitting.** **a** Bias-dependent fluorescence spectra of a single QD (QD A) under TPE. The range of $X$ ($XX$) photon energy tuned by the quantum-confined Stark effect is 1.08 meV (0.76 meV). Arrows mark photon energies where the tuning of FSS is demonstrated in **c**–**h**. **b** Polarization dependence of the energy difference between $XX$ and $X$ peaks shown in **a** at $V_g = 0.1$ V. Error bars indicate Gaussian fitting residual standard error. Fitting (red) with a sine function gives an FSS of 2.92(7) $\mu$eV. **c**–**h** Elimination of FSS at various biases via the AC Stark effect. Error bars indicate the standard error of sine fit residuals.

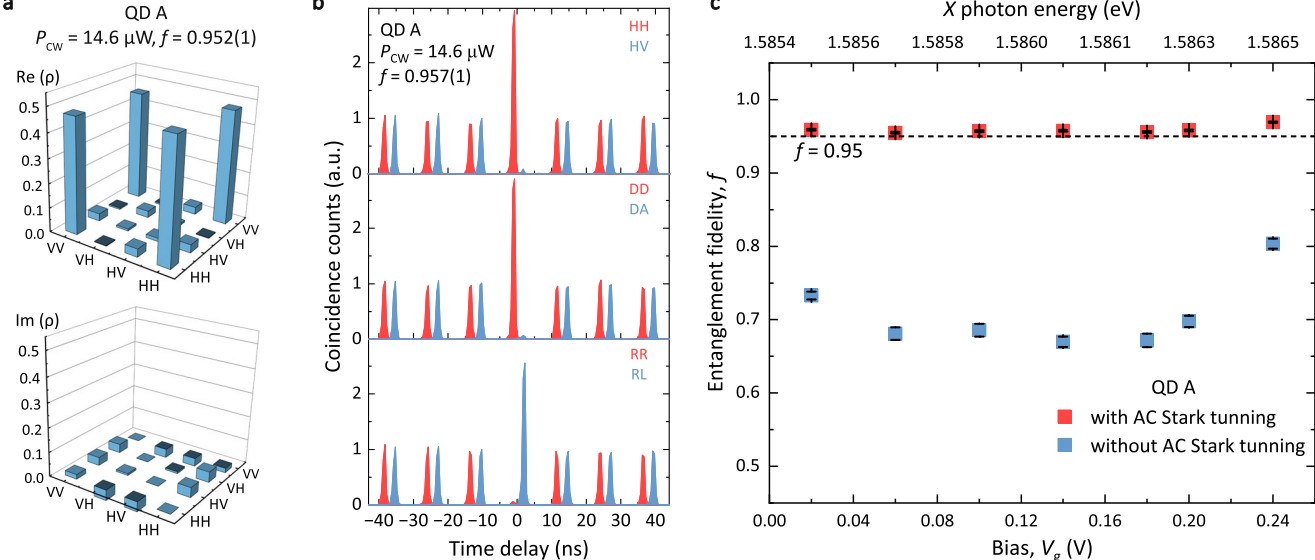

**Fig. 3 | Measurement of two-photon entanglement fidelity of QD A.** **a** Real and imaginary part of the two-photon density matrix obtained by 16 cross-correlation measurements, giving an entanglement fidelity with respect to the state $|\Phi^+\rangle$ of $f = 0.952(1)$. The FSS is compensated by a CW laser with $\hbar\delta_{CW} = 303$ $\mu$eV and power $P_{CW} = 14.6$ $\mu$W at $V_g = 0.10$ V. **b** Measurement of the entanglement fidelity with a reduced measurement basis: linear (top), diagonal (middle) and circular (bottom). $f = 0.957(1)$. The peaks for cross-polarizations (blue) are shifted by 3 ns for clarity. **c** Entanglement fidelity as a function of bias. Blue (red) squares: measured entanglement fidelity before (after) eliminating the FSS by AC Stark effect.

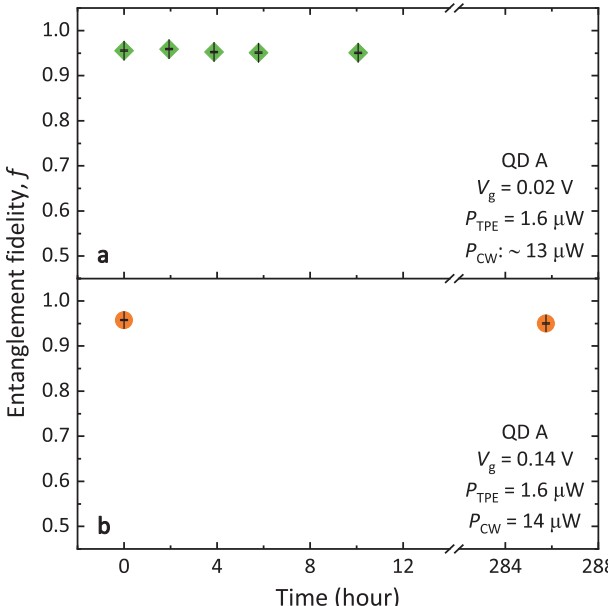

power. To examine the stability of our EPS and hybrid tuning scheme, we measure $f$ in different time scales. Figure 4a shows $f$ continuously monitored for 10 hours without any close-loop control under the condition of $V_g = 0.02$ V, $P_{TPE} = 1.6$ $\mu$W, and $P_{CW}$ in the range of 12 to 13.5 $\mu$W. Additionally, no optimization of the sample position or polarization correction is performed within 10 hours. During this period, $f$ is evaluated every 2 hours, yielding an average $f$ of 0.954 with a standard deviation of 0.004, demonstrating the continuous operation of the EPS in a time scale of hours.

For an even longer term, we compare $f$ measured with an interval of 11.9 days at $V_g = 0.14$ V, $P_{TPE} = 1.6$ $\mu$W and $P_{CW} = 14$ $\mu$W (see Fig. 4b). Without re-adjusting the bias and laser power, $f$ remains almost the same (0.958(1) and 0.950(1), respectively), exhibiting an excellent stability of our EPS in a time scale of days.

## Scalability

In previous sections, we demonstrated a robust wavelength-tunable EPS using a QD with a relatively small FSS (2.92(7) $\mu$eV). From an application standpoint, it is important to show the ability to tune multiple QDs into resonance while maintaining a high entanglement fidelity. In order to verify the scalability of the hybrid tuning scheme, we characterize 344 QDs (Fig. 5a), revealing an average wavelength tuning range of 1.27(31) meV (Fig. 5b) and an average FSS of 7.92(364) $\mu$eV (Fig. 5c). We categorize them into groups represented by bars in Fig. 5a where all QDs can be tuned into resonance. From this statistical analysis, one can easily find several groups containing tens of QDs each. Bias maps for a representative group of up to 39 QDs

**Fig. 4 | Stability. a** Entanglement fidelity $f$ of QD A continuously measured at $V_g = 0.02$ V with TPE laser power ($P_{TPE}$) of 1.6 $\mu$W and CW laser power ($P_{CW}$) of around 13 $\mu$W for 10 hours. **b** $f$ measured with an interval of 11.9 days at $V_g = 0.14$ V, $P_{TPE} = 1.6$ $\mu$W and $P_{CW} = 14$ $\mu$W.

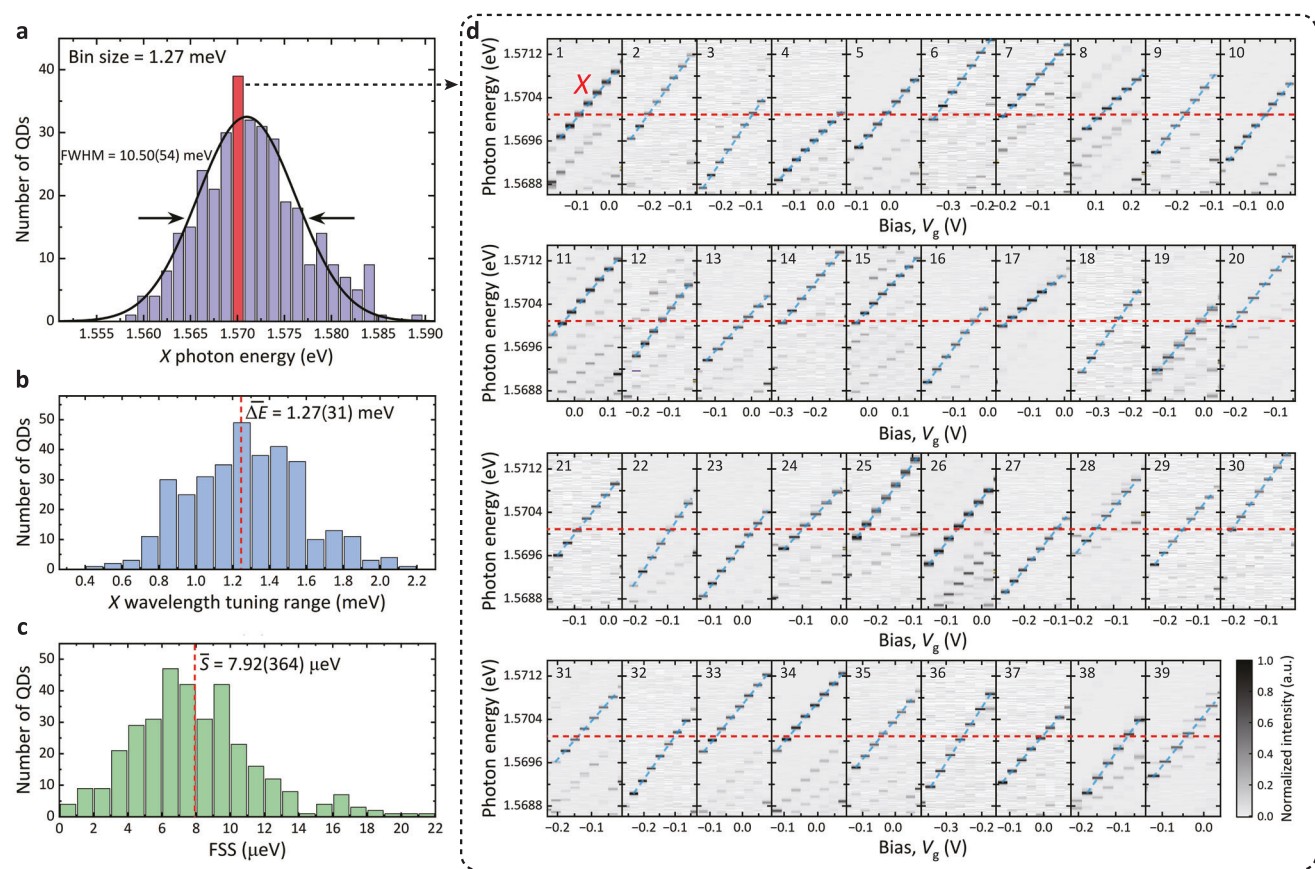

**Fig. 5 | Statistical analysis of QDs in resonance. a** $X$ photon energy distribution of 344 randomly selected QDs in a single chip. The bin size of each bar is 1.27 meV equal to the average value of the $X$ wavelength tuning range for the measured QDs. Black line: fitting with a Gaussian function, yielding an inhomogeneous broadening with a full width at half maximum (FWHM) of 10.50(54) meV. **b** Distribution of wavelength tuning range. Red dashed line: the average value of tuning range

($\overline{\Delta E} = 1.27(31)$ meV). **c** Distribution of FSS. Red dashed line: the average value of FSS ($\overline{S} = 7.92(364)$ $\mu$eV). **d** Bias-dependent fluorescence spectra of the $X$ state for 39 QDs in the group marked by the red bar in **a**. Wavelength tuning ranges, indicated by blue dashed lines, of all QDs intersect 1.5701 eV (marked by red dashed lines), indicating these 39 QDs can be tuned into resonance.

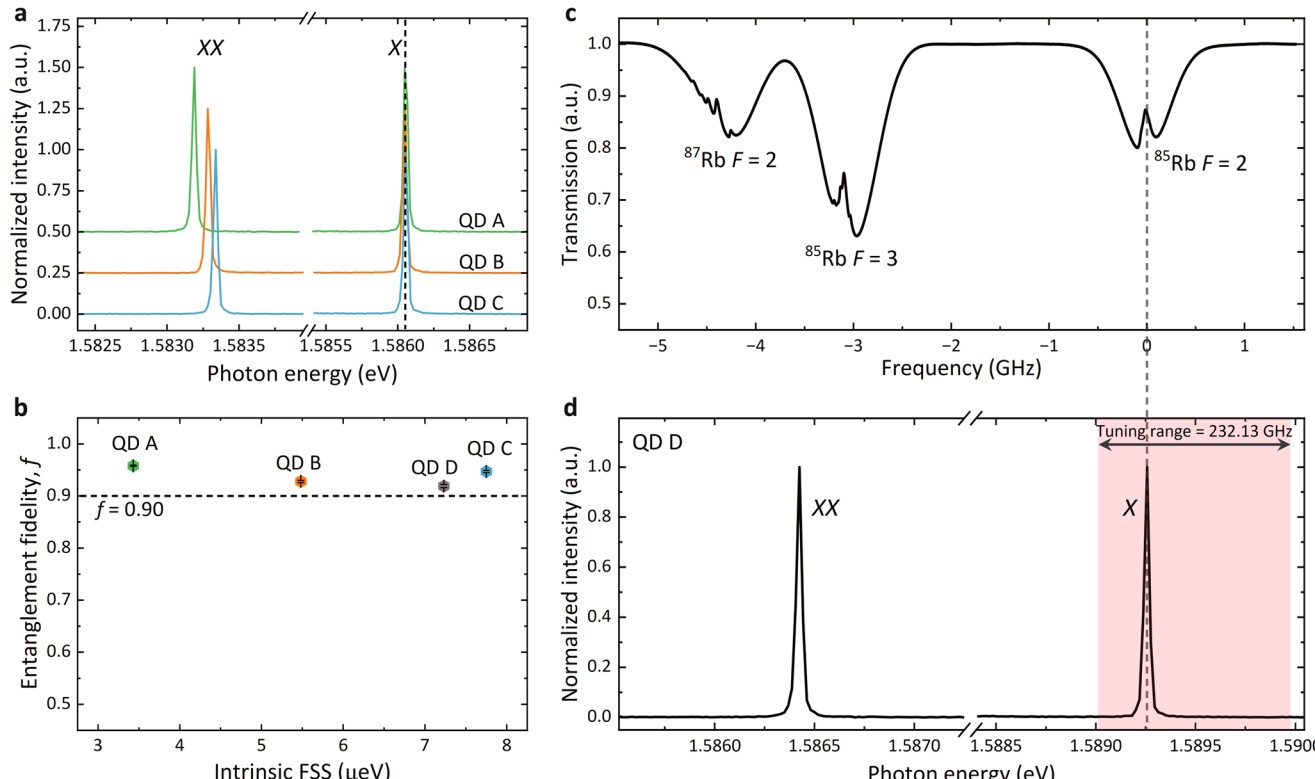

**Fig. 6 | Multiple wavelength-matched QD EPSs with high entanglement fidelity.**
**a** Fluorescence spectra measured when the $X$ of three QDs are tuned to resonance. Black dashed line: guide for the eye. **b** Entanglement fidelity for each QD after eliminating FSS. **c** Saturated absorption spectrum of Rb transition from $5S_{1/2}$ to $5P_{3/2}$ measured with a laser at room temperature. **d** Fluorescence spectrum of QD D, where $X$ is tuned to resonate with the saturated absorption signal of $^{85}$ Rb $F = 2$ (gray dashed line). The red-shaded area indicates the $X$ wavelength tuning range of 232.13 GHz for QD D.

(highlighted by the red bar in Fig. 5a) are depicted in Fig. 5d. Tuning ranges, indicated by blue dashed lines, of all QDs in this group intersect 1.5701 eV marked by red dashed lines, proving that all QDs can be tuned to the same emission wavelength. These results exhibit the excellent scalability of our scheme for wavelength matching in multiple QDs.

Following the demonstration of tuning multiple QDs into resonance, it is crucial to verify if high entanglement fidelity is still attainable under these conditions. To this end, we tune three QDs (labeled A, B, and C) into resonance (Fig. 6a) and measure their entanglement fidelity after FSS is eliminated (see Supplementary Fig. 5 for details). A fidelity above 0.9 has been observed with all three QDs (A: 0.958(1), B: 0.928(2), and C: 0.947(2)) as shown in Fig. 6b, confirming that multiple EPSs with the same emission wavelength and high entanglement fidelity can be achieved simultaneously.

Furthermore, a fully functional quantum repeater requires not only EPSs but also quantum memories. One of the advantages of GaAs QDs used in this work is that they emit at around 780 nm, matching emission lines of Rb atoms which are one of the most promising candidates for quantum memories[49–54]. To illustrate the feasibility of interfacing our EPS with Rb atoms, we tune a QD (labelled QD D, see Fig. 6d) in resonance with the D2 line of $^{85}$ Rb $F = 2$ (see the saturated absorption spectrum[55] in Fig. 6c). Again, a high entanglement fidelity of 0.919(3) is obtained with QD D (see Fig. 6b).

## Discussion

The hybrid tuning scheme demonstrated in this work has the following advantages: 1. It is compatible with various micro/nanophotonic structures, such as micropillar cavities[56], open cavities[57], bull's eye cavities[58,59] and photonic crystal structures[60,61], that are needed to improve the brightness, photon indistinguishability, and

entanglement fidelity. 2. Combining with electrically isolated nanostructures[62], this scheme in principle allows local tuning of different QDs on the same chip, essential for integrated quantum optical circuits with multiple EPSs[63].

Despite these advantages, the performance of the device and tuning scheme can be further optimized. Firstly, the number of QDs that can be tuned into resonance could be increased either by reducing the intrinsic non-uniformity of QD emission wavelength or by extending the wavelength tuning range. The former can potentially be realized by increasing the QD size via deeper etching (see Supplementary Fig. 6). Meanwhile, the latter can be achieved by increasing the thickness and height of the tunnel barriers. A tuning range up to 25 meV has been reported with InGaAs QD embedded in $Al_{0.75}Ga_{0.25}As$ barriers[27,29]. Secondly, QDs with relatively large FSS require high CW laser power to fully compensate the FSS. Filtering the CW laser for these QDs could be quite challenging considering the small detuning between the CW laser and the biexciton photon. This potential problem can be solved by integrating QDs into micro/ nano cavities[56–58,60] where the light-matter interaction is significantly enhanced and the requirement for the CW laser power is much lower.

In conclusion, we have demonstrated a wavelength-tunable QD EPS with entanglement fidelity above 0.955(1) by applying a hybrid tuning scheme to a droplet-etched GaAs QD. This scheme combines AC and quantum-confined Stark effects, which enables simultaneous tuning of exciton FSS and QD emission wavelength. With this tuning scheme, our device can be operated continuously over a long period of time without the need for re-calibration. QD statistics measurements demonstrate the ability to tune multiple QDs into resonance while persevering an entanglement fidelity above 0.9. In addition to QDs, our tuning scheme is also applicable to a variety of deterministic EPSs, such

as perovskite/II–VI colloidal nanocrystals[64–66] and quantum emitters in 2D materials[67,68]. Our work makes an important step towards robust and scalable on-demand EPSs for quantum internet and integrated quantum optical circuits.

## Methods

### Sample preparation

The sample is fabricated following the methodology developed for charge-tunable GaAs QDs[69]. The sample structure can be found in Supplementary Note 1. We highlight the most relevant part, the growth of the local droplet etched GaAs QDs: On an AlGaAs surface at a pyrometer reading of 560 °C and an arsenic equivalent pressure (BEP) of $4.5 \times 10^{-7}$ Torr, nominally 0.31 nm Al is deposited. After 120 s, nanoholes are formed on the surface. This etching procedure is stopped by restoring the As-flux to a standard III–V-growth BEP of $9.6 \times 10^{-6}$ Torr. Finally, the etched nanoholes were filled with 1.0 nm GaAs to form QDs. The structure is finalized by AlGaAs layers.

### Measurement techniques

The QD sample is placed in a closed-loop cryostat (attoDRY1000, $T$ = 3.6 K). For two-photon resonance excitation, a laser pulse with a duration of 140 fs generated by a Ti-sapphire laser with a repetition rate of 80 MHz is stretched to ~6 ps by a homemade pulse shaper[70]. For tuning FSS, a tunable narrow-linewidth CW laser (Toptica 780DL pro) is used. The CW and pulse laser are primarily suppressed by four tunable notch filters. Any remaining laser background is further removed by a grating-based filter (see Supplementary Fig. 7). A spectrometer with a focal length of 750 mm (Princeton Instruments, HRS-750) is used to acquire fluorescence spectra. For cross-correlation measurements, $X$ and $XX$ are separated by a volume phase holographic transmission grating and detected by two single-photon avalanche diodes (SPAD) with a time resolution ~400 ps. A quarter-wave plate (QWP), a HWP, and a polarizer are placed in front of the SPAD to set different polarization bases. In full quantum state tomography, the two-photon density matrix $\rho$ is reconstructed by 16 cross-correlation measurements between $X$ and $XX$ photons with the maximum likelihood method according to ref. 45. The value of entanglement fidelity is obtained by $f = \langle \Phi^+ | \rho | \Phi^+ \rangle$. In a reduced set of projective measurements, the entanglement fidelity is extracted by 6 cross-correlation measurements (see details in Supplementary Note 4).

## Data availability

The raw data that support the findings of this study are available at https://doi.org/10.5281/zenodo.11402438 and from the corresponding author upon request.

## Code availability

The codes that have been used for this study are available from the corresponding author upon request.

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

## Acknowledgements

We thank Nadine Viteritti for the electrical contact preparation of the sample. We thank Prof. Zhao-Ying Wang and Dr. Ying Wang for fruitful discussions. We thank Qian-Yi Chen and Fang-Yuan Li for their help with supplementary experiments. F.L. acknowledges support from the National Key Research and Development Program of China (No. 2023YFB2806000, 2022YFA1204700) and the National Natural Science Foundation of China (U21A6006, 62075194, 61975177, U20A20164, 11934011, 12325412). H.-G.B., A.D.W., and A.L. acknowledge support by the BMBF-QR.X Project 16KISQ009 and the DFH/UFA, Project CDFA-05-06.

## Author contributions

F.L., J.-Y.Y. and C.C. conceived the project. H.-G.B., C.H., A.D.W. and A.L. grew the wafer, made the AFM measurements and fabricated the sample. C.C. and J.-Y.Y. constructed the optical setup and carried out the experiments. C.C., J.-Y.Y. and F.L. analysed the data. J.W., X.X. and H.C. provided support for the measurement of saturated absorption spectrum. D.-W.W. and J.Z. provided theoretical support. X.L., Q.Y., W.F., R.-Z.L., Y.-H.H., W.E.I.S., C.-Y.J. and F.L. provided supervision and expertise. C.C., J.-Y.Y. and F.L. wrote the manuscript with comments and inputs from all the authors.

## Competing interests

The authors declare no competing interests.
