## [Peer Review File · Nature Communications]

Wavelength-tunable high-fidelity entangled photon sources enabled by dual Stark effectsREVIEWERS' COMMENTS:

Reviewer #1 (Remarks to the Author):

Dear Editor,

The manuscript entitled “Wavelength-tunable high-fidelity entangled photon sources enabled by dual Stark effects” shows very nice results. What is particularly interesting is that the authors seem to control the AC Stark effect better than previous groups, so to have many dots on the same samples to which they can apply their correction schemes.

The combination of both conventional electric field induced Stark effect and the AC one is not per se “novel” or a unique idea. In many ways it is an obvious thing to do. Here the authors manage to show that the scheme works on very many dots, that the entanglement fidelity is anyway extremely high, and in general all seems really good.

As a reviewer I am impressed that things work so well. So I can only suggest the manuscript should be published ASAP.

I have a few suggestions as minor comments:

-maybe the author could comment on what is it that makes things so easy for them? Maybe there is a small mistake others did in the literature, making the scheme only a success for some few specific cases? And now they have a more universal scheme?

-the authors should tone down the repeater story, as also indistinguishability is needed for that, and they did not prove it in this paper, even if they claim very narrow lines and their dots seem of very high quality.

-The authors should describe in details (layer by layer thickness, doping levels) their sample structure. They here give a reference to a paper which also references another one on that..and the reader should not be sent to a sample structure hunt through the literature.... the authors should simply put in the supplementary materials the full detailed sample structure, as that is probably part of the success story.

Reviewer #2 (Remarks to the Author):

The authors report the demonstration of wavelength-tunable entangled photon sources based on droplet-etched GaAs/AlGaAs quantum dots. The combined use of Optical and quantum-confined Stark effects permits them to obtain a sizeable reduction of the fine

structure splitting of the X-XX cascade together with a limited emission tuning (around 1 meV). Both methods, the optical Stark effect for the reduction of the fine structure splitting and the quantum confined Stark effect, are known from decades and have been applied to the tuning of the QD emission properties. Here the novelty is to apply them at the same time. However, the results are, at the moment, quite limited, especially in the wavelength tuning range. The authors suggest several ways to improve such shortcomings in the discussion section (which is more a perspective rather than a discussion section), but they did not try any.

The paper is well written, with a complete set of references. The data are sound and well documented in the paper and the supplement.

Reviewer #3 (Remarks to the Author):

Chen et al. report on an approach to realize a wavelength-tunable source of entangled photons. The authors use droplet-etched GaAs QDs, which have been shown to be a high-quality entangled photon source in prior work. The primary advance in the present paper is the simultaneous tuning of the biexciton wavelength and FSS using the quantum-confined Stark effect and AC Stark effect, respectively, while maintaining the entanglement fidelity. Both tuning approaches have been utilized in previous work, as referenced by the authors. This dual tuning capability is important for the scalability of the entangled photon source, and the authors demonstrate their approach on numerous QDs while maintaining an entanglement fidelity above 0.955. I find that the authors provide appropriate experimental support and analysis for the results that they present. However, because only a basic proof-of-concept is shown using tuning techniques that have been established in prior work, I do not find the paper to be suitable for Nature Communications in its present form. The following comments need to be addressed:

- One of the author's stated objectives is to overcome the non-uniformity of GaAs QD emission wavelengths and FSS. Although demonstrating the basic proof-of-concept for tuning the wavelength and FSS of 28 QDs is important, no data is shown on wavelength-matched GaAs QDs. This is an essential demonstration to support the author's claims that

their approach can produce a scalable source of entangled photons that could be used as part of a quantum repeater. Can high fidelity be maintained for two or more QDs that have been tuned into mutual resonance? This would be a better measure of the scalability of this approach and would make a stronger case for the utility of their approach.

- Additionally, while not essential, the level and scope of this paper could be raised by tuning a QD into resonance with Rb atoms emission line.

- The authors characterize the stability of the entanglement fidelity over a period of 10 hours and nearly 12 days, claiming that the experimental conditions were unchanged during this time. If I understand correctly, does this mean there was no correction for drift (e.g., due to small changes in the spatial position, polarization conditions due to changes due to small changes in the ambient environment, etc)? The authors should be more specific with what they mean by “without the need for re-calibration”.

- The authors demonstrate the scalability of their approach with results from 28 QDs, noting that the wavelength tuning range of 93% of the QDs overlap with at least one other QD. For scaling in integrated photonics systems this is quite low, essentially limiting devices to about two wavelength-matched sources. Can the authors comment on this limitation, and the feasibility of extending the tuning range with this approach?

Lines 154, 282: Typos/grammar

Response to reviewers' comments:

Reviewer #1 (Comments for the Author):

R1.1: The manuscript entitled "Wavelength-tunable high-fidelity entangled photon sources enabled by dual Stark effects" shows very nice results. What is particularly interesting is that the authors seems to control the AC Stark effect better than previous groups, so to have many dots on the same samples to which they can apply their correction schemes.

The combination of both conventional electric field induced stark effect and the AC one is not per se "novel" or a unique idea. In many ways it is an obvious thing to do. Here the authors manage to show that the scheme works on very many dots, that the entanglement fidelity is anyway extremely high, and in general all seems really good.

As a reviewer I am impressed that things work so well. So I can only suggest the manuscript should be published ASAP.

A1.1: We sincerely thank the reviewer's highly positive comments.

R1.2: maybe the author could comment on what is it that makes things so easy for them? Maybe there is a small mistake others did in the literature, making the scheme only a success for some few specific cases? And now they have a more universal scheme?

A1.2: We thank the reviewer for appreciating our work. Compared with previous works, there are two main reasons that lead to the successful implementation of this hybrid tuning scheme to many quantum dots and obtain high entanglement fidelity.

Firstly, the quality of the QD wafer is highly improved due to the advanced growth technique and the integration of n-i-p diode. Compared to previous studies, the nearly defect-free crystal quality and extremely low charge noise of our droplet-etched GaAs QDs, as evidenced by blinking-free emission and near transform-limited linewidth (Nat. Nanotechnol. 18, 1139), providing the foundation for attaining near-unity entanglement fidelity. In contrast, QDs used in past works might not achieve perfect entanglement fidelity due to spectral diffusion caused by charge noise, even after minimizing FSS. To further clarify details of the sample, we have added a section to describe the wafer structure in Supplementary Information Section 1.

Secondly, two-photon resonant excitation method is used. In the previous work (Phys. Rev. Lett. 103, 217402), although the FSS was eliminate via AC Stark effect, non-resonant above-barrier excitation was used to generate biexcitons, leading to stronger charge noise and hence lower entanglement fidelity. In contrast, our work employs resonant two-photon resonant excitation to deterministically create biexcitons, effectively reducing laser-induced charge noise and improving entanglement fidelity.

R1.3: the authors should tone down the repeater story, as also indistinguishability is needed for that, and they did not prove it in this paper, even if they claim very narrow lines and their dots seem of very high quality.

A1.3: We agree with the reviewer that photon indistinguishability is another figure of

merit for the ultimate implementation of our EPSs in a quantum repeater.

Following the reviewer's suggestion, we have minimized the use of the term "quantum repeater" in the abstract and introduction. Additionally, we have strengthened the connection between our device and future quantum repeater applications by demonstrating the ability to tune our entangled photon sources into resonance with Rb atoms, a leading candidate for quantum memories. This result evidences the potential for a feasible quantum repeater employing QD EPSs interfaced with Rb atom memories.

R1.4: The authors should describe in details (layer by layer thickness, doping levels) their sample structure. They here give a reference to a paper which also references another one on that..and the reader should not be sent to a sample structure hunt through the literature.... the authors should simply put in the supplementary materials the full detailed sample structure, as that is probably part of the success story.

A1.4: We completely agree with the reviewer that a detailed sample structure is necessary. We have added a section (Section I) in Supplementary Information to provide a detailed description of the sample:

The heterostructure is grown by molecular beam epitaxy (MBE) technology on a [001]-oriented GaAs substrate. Quantum dots are embedded in an n-i-p diode structure, enabling effective manipulation of both the electric field experienced by the QDs and their charge states. The n-contact consists of Si-doped $\text{Al}_{0.15}\text{Ga}_{0.85}\text{As}$ with a doping concentration of $2 \times 10^{18} \text{ cm}^{-3}$. A 20 nm $\text{Al}_{0.15}\text{Ga}_{0.85}\text{As}$ layer and a 10 nm $\text{Al}_{0.33}\text{Ga}_{0.67}\text{As}$ layer serve as a tunnel barrier to separate the QDs from the n-contact. GaAs quantum dots are grown in the $\text{Al}_{0.33}\text{Ga}_{0.67}\text{As}$ layer using the local droplet etching method, followed by a 273.6 nm $\text{Al}_{0.33}\text{Ga}_{0.67}\text{As}$ layer as the blocking barrier. The p-contact consists of 65 nm C-doped $\text{Al}_{0.15}\text{Ga}_{0.85}\text{As}$ (p+, doping concentration of $2 \times 10^{18} \text{ cm}^{-3}$), 10 nm C-doped $\text{Al}_{0.15}\text{Ga}_{0.85}\text{As}$ (p++, doping concentration of $8 \times 10^{18} \text{ cm}^{-3}$), and 5 nm C-doped GaAs (p++, doping concentration of $8 \times 10^{18} \text{ cm}^{-3}$). Below the n-i-p diode structure, a distributed Bragg reflector consisting of 10 pairs of AlAs (67.08 nm thick)/ $\text{Al}_{0.33}\text{Ga}_{0.67}\text{As}$ (59.54 nm thick) is grown to enhance the collection efficiency of photons. The overall structure is illustrated in Supplementary Fig.1.

Reviewer #2:

R2.1: The authors report the demonstration of wavelength-tunable entangled photon sources based on droplet-etched GaAs/AlGaAs quantum dots. The combined use of Optical and quantum-confined Stark effects permits them to obtain a sizeable reduction of the fine structure splitting of the X-XX cascade together with a limited emission tuning (around 1 meV). Both methods, the optical Stark effect for the reduction of the fine structure splitting and the quantum confined Stark effect, are known from decades and have been applied to the tuning of the QD emission properties. Here the novelty is to apply them at the same time. However, the results are, at the moment, quite limited, especially in the wavelength tuning range. The authors suggest several ways to improve such shortcomings in the discussion section (which is more a perspective rather than a discussion section), but they did not try any.

A2.1: We appreciate the reviewer’s thoughtful comment. In response, we have undertaken extensive new experiments and significantly revised the manuscript to address these concerns. We believe the new data strengthens our claims, improves the demonstration of our hybrid tuning scheme, and broadens the scope of the work. The new data and revisions to the manuscript are summarized below:

1. Improved tuning range: Although it is mentioned in the previous manuscript that the wavelength tuning range can be further improved, we would like to stress that the tuning range of the current device is already sufficiently large to tune up to **39 QDs** into resonance, which, to our best knowledge, has never been demonstrated before. To showcase our tuning capability more comprehensively, we characterized 344 QDs on a single chip. This measurement reveals a maximum tuning range of 2.1 meV (see RFig. 1 below), about two times of the tuning range (~ 1 meV) mentioned in the last manuscript. Furthermore, we categorized the QDs into groups capable of achieving resonance, revealing several groups containing tens of wavelength-matched QDs. This statistical analysis has been presented in Fig. 5 in the revised manuscript.

RFig. 1. X photon tuning range (2.1 meV) of a QD. Dashed line: guide for the eye.

2. Wider wavelength tuning range can be achieved by increasing the thickness and height of the tunnel barrier. The feasibility of this approach has been confirmed in the literature (Appl. Phys. Lett. 97, 031104 and Nat. Phys. 6, 947). Along this direction, we have been investigating new QD growth parameters, obtaining a reduced inhomogeneous broadening (FWHM = 3.97 meV, see RFig. 2) in characterizing 794 QDs. This data shown in RFig. 2 has also been added to the Supplementary Information (Supplementary Fig. 5). The improved uniformity together with the potential for a larger tuning range, suggests an even higher number of achievable wavelength-matched QDs. While the fabrication of fully functional new devices requires further development and extensive time, our preliminary results and previous literature provide a solid foundation for the successful implementation of our proposed approach.

RFig. 2. X photon energy distribution of 794 randomly selected quantum dots grown with a new set of parameters.

3. Demonstration of multiple wavelength-matched EPSs with high entanglement fidelity: We demonstrate the capability of tuning multiple QDs into resonance while maintaining high entanglement fidelity. We tune three QDs into resonance and observed high entanglement fidelity (0.957(1), 0.928(2) and 0.947(2)) with all of them after eliminating FSS (see Figs. 6a, b in the revised manuscript).

4. Demonstration of a QD EPS resonant with Rb atoms: Building a fully functional quantum repeater requires not only EPSs but also quantum memories. In order to show the potential of interfacing our EPS with Rb atoms, one of most promising candidates of quantum memories, we tune a QD in resonance with the D2 line of Rb atoms (see the saturated absorption spectrum in Fig. 6c). A high entanglement fidelity of 0.919(3) is obtained with this QD after FSS tuning. These results are presented in Figs. 6c, d of the revised manuscript.

R2.2: The paper is well written, with a complete set of references. The data are sound and well documented in the paper and the supplement.

A2.2: We thank the reviewer again for appreciating our work.

Reviewer #3 (Comments for the Author):

R3.1: Chen et al. report on an approach to realize a wavelength-tunable source of entangled photons. The authors use droplet-etched GaAs QDs, which have been shown to be a high-quality entangled photon source in prior work. The primary advance in the present paper is the simultaneous tuning of the biexciton wavelength and FSS using the quantum-confined Stark effect and AC Stark effect, respectively, while maintaining the entanglement fidelity. Both tuning approaches have been utilized in previous work, as referenced by the authors. This dual tuning capability is important for the scalability of the entangled photon source, and the authors demonstrate their approach on numerous QDs while maintaining an entanglement fidelity above 0.955. I find that the authors provide appropriate experimental support and analysis for the results that they present. However, because only a basic proof-of-concept is shown using tuning techniques that have been established in prior work, I do not find the paper to be suitable for Nature Communications in its present form. The following comments need to be addressed:

One of the author's stated objectives is to overcome the non-uniformity of GaAs QD emission wavelengths and FSS. Although demonstrating the basic proof-of-concept for tuning the wavelength and FSS of 28 QDs is important, no data is shown on wavelength-matched GaAs QDs. This is an essential demonstration to support the author's claims that their approach can produce a scalable source of entangled photons that could be used as part of a quantum repeater. Can high fidelity be maintained for two or more QDs that have been tuned into mutual resonance? This would be a better measure of the scalability of this approach and would make a stronger case for the utility of their approach.

A3.1: We thank the reviewer for the insightful comment. We fully agree that demonstrating wavelength-matched QDs with high entanglement fidelity is crucial to support our claims and better reflects the potential of our approach.

Stimulated by reviewer's suggestion, we tuned 3 QDs into resonance and measured their entanglement fidelities (see Figs. 6a, b in the revised manuscript). Notably, all three QDs retained high entanglement fidelities: 0.957(1), 0.928(2), and 0.947(2). These results unambiguously demonstrate the capability of our hybrid tuning scheme to achieve on-demand, wavelength-matched EPSs.

R3.2: Additionally, while not essential, the level and scope of this paper could be raised by tuning a QD into resonance with Rb atoms emission line.

A3.2: We thank the reviewer for this valuable suggestion. As suggested, in the revised manuscript, we demonstrate a QD with high entanglement fidelity ($f=0.919(3)$) and emission wavelength resonant with the D2 line of Rb atom (see Figs. 6c, d). Fig.6c shows the saturated absorption spectrum of Rb atoms transitioning from $5S_{1/2}$ to $5P_{3/2}$ at room temperature. In Fig. 6d, we show the fluorescence spectrum of a QD with the X emission wavelength tuned to be resonant with the ^{85}Rb $F = 2$. The entanglement fidelity measured under this condition yields $f = 0.919(3)$ (see Figs. 6b). Moreover, it is

worth mentioning that the X wavelength tuning range of this QD is 0.96 meV (232.13 GHz), enabling resonance with other absorption peaks of the D2 line of Rb atoms.

R3.3: The authors characterize the stability of the entanglement fidelity over a period of 10 hours and nearly 12 days, claiming that the experimental conditions were unchanged during this time. If I understand correctly, does this mean there was no correction for drift (e.g., due to small changes in the spatial position, polarization conditions due to changes due to small changes in the ambient environment, etc)? The authors should be more specific with what they mean by “without the need for re-calibration”.

A3.3: We appreciate the reviewer’s careful review and acknowledge the need for clarity regarding our description of long-term stability. The term “without the need for re-calibration or re-adjustment” primarily refers to two key parameters: **the bias applied to the sample and the laser power** which controls the emission wavelength and FSS, respectively. Additionally, during the 10-hour stability test, we did not perform any adjustments to the spatial position or polarization. For the extended 11.9-day test, the sample position and polarization condition were re-optimized due to other intervening experiments performed on the same setup.

To avoid any ambiguity, we have explicitly modified the manuscript to clarify that “re-calibration or re-adjustment” solely refers to the bias voltage and laser power. This revised statements have been added to the following sections of the manuscript:

The stability of the QD EPS is confirmed by maintaining high entanglement fidelity over a long period of time without re-adjustment of the bias and laser power.

...ideally without the concern of periodic re-adjustment of the bias and laser power...Additionally, no optimization of the sample position or polarization correction are performed within 10 hours...

For an even longer term, we compare f measured with an interval of 11.9 days at $V_g = 0.14$ V, $P_{TPE} = 1.6$ μ W and $P_{CW} = 14$ μ W (see Fig. 4b). Without re-adjusting the bias and laser power, f remains almost the same (0.958(1) and 0.950(1), respectively...

R3.4: The authors demonstrate the scalability of their approach with results from 28 QDs, noting that the wavelength tuning range of 93% of the QDs overlap with at least one other QD. For scaling in integrated photonics systems this is quite low, essentially limiting devices to about two wavelength-matched sources. Can the authors comment on this limitation, and the feasibility of extending the tuning range with this approach?

A3.4: We thank the reviewer for pointing out this issue. Our description and presentation of the data in the scalability section in the previous manuscript may give readers an impression that only two wavelength-matched sources can be found on this device. In fact, if we draw a horizontal line in the old Fig. 5b (see the figure below),

this line would intersect with up to seven QDs most, indicating seven QDs can be tuned into resonance.

RFig. 2. (The original Fig. 5b in the previous manuscript). Tuning range of X photon energy of 28 QDs. Up to 7 QDs intersect with a horizontal line representing constant wavelength, indicating 7 QDs can be tuned into resonance.

From the reviewer's comment, we realize that we did not sufficiently communicate this important aspect of our work. To better demonstrate the scalability of our approach, we have undertaken an extensive characterization of 344 QDs and categorized them into groups capable of achieving resonance. From this statistical analysis, it is easy to find several groups containing tens of wavelength-matched QDs. Bias maps for a representative group of up to 39 QDs are depicted in Fig. 5d in the revised manuscript (see also the figure below). Tuning ranges of all QDs in this group intersect 1.5701 eV, proving that all 39 QDs can be tuned to the same emission wavelength. These results more clearly exhibit the scalability of our scheme for wavelength matching in multiple QDs.

RFig. 3. (Fig. 5d in the revised manuscript). Bias-dependent fluorescence spectra of the neutral exciton state for 39 QDs. Wavelength tuning ranges, indicated by blue dashed lines, of all QDs intersect 1.5701 eV (marked by red dashed lines), indicating these 39 QDs can be tuned into resonance.

The wavelength tuning range can be extended by increasing the thickness and height of the tunnel barrier. A tuning range up to 25 meV has been reported with InGaAs QD embedded in $\text{Al}_{0.75}\text{Ga}_{0.25}\text{As}$ barriers (Appl. Phys. Lett. 97, 031104 and Nat. Phys. 6, 947), confirming the feasibility of this approach. Additionally, improving the intrinsic uniformity of QD emission wavelength would also increase the number of wavelength-matched QDs. We have started to explore new QD growth parameters. Our preliminary results show a reduced inhomogeneous broadening (FWHM = 3.97 meV, see Supplementary Fig. 5). These contents have been added to the Discussion section in the revised manuscript.

R3.5: Lines 154, 282: Typos/grammar

A3.5: We thank the reviewer for pointing out these mistakes. The manuscript is revised as follows:

1. Line 154: horizontally, polarized CW laser slightly red-detuned from $|XX\rangle \rightarrow |X_H\rangle$

transition shifts X_H energy while **leaving** the cross-polarized X_V state unchanged (see Fig. 1d).

2. Since Fig.5 is completely replaced by a new set of data, the sentence in the original line 282 has been deleted.

REVIEWERS' COMMENTS

Reviewer #1 (Remarks to the Author):

The authors improved their manuscript from the original version, basically showing that their approach applies to many QDs, and that they can tune their dot emission in resonance to atomic transitions, which would offer an alternative path for generating indistinguishability in a "complete" system approach.

In general I had a positive opinion about the first version, and I can only see this version as an improvement on the first one. I think that the minor comments I raised have been addressed, and also there are strong arguments addressing the other reviewer's comments. Personally I do not think that the fact that individually the techniques used are not original reduce the value of this contribution. Effectively they show results that have not been obtained by others in terms of tuneability and repeatability of fidelity etc., to a level which I think deserves publication and recognition from the community.

Maybe worth noting, their contribution in this version is also one of the few discussing uniformity issues associated to droplet epitaxy dots (which are very uniform indeed, at least in some instances, but whose uniformity is surprisingly little discussed). I think this is also a value, even if not the main reason for publication obviously.

Reviewer #2 (Remarks to the Author):

I find the authors have answered to all the criticisms made by the referees, including myself. I think that the authors have deeply improved the manuscript adding several new data sets and new figures. I really appreciate the better statistical analysis of the tuning capabilities of the system, while maintaining a low FSS by AC Stark effect and the demonstration of the possibility to tune the QDs in a useful wavelength range.

Reviewer #3 (Remarks to the Author):

The authors have responded to all referee comments and have also performed additional measurements. In response to my concerns about scalability, the authors have tuned three QDs into resonance while maintaining entanglement fidelity. These results are shown in Fig.

6. Similarly, they also provide a demonstration of tuning a QD into resonance with the D2 line of the Rb atom. Both of these additions raise the level of the paper. However, it would be helpful to show tuning steps in both cases (possibly in the supplement or as an inset to Figs. 6a/d).

I find that the additional data and clarifications provided in response to all referee comments have addressed my concerns and have raised the level of the paper. I recommend this manuscript for publication in Nature Communications.